

# Properties evaluation of silorane, low-shrinkage, non-flowable and flowable resin-based composites in dentistry

Rodrigo R. Maia[1], Rodrigo S. Reis[2], André F.V. Moro[3], Cesar R. Perez[3], Bárbara M. Pessôa[3] and Katia R.H.C. Dias[4]

[1] Department of Operative Dentistry, University of Iowa, Iowa City, USA
[2] Pontifical Catholic University, Rio de Janeiro, RJ, Brazil
[3] Department of Dentistry, State University of Rio de Janeiro, Rio de Janeiro, RJ, Brazil
[4] Department of Dental Clinic, Federal University of Rio de Janeiro, Rio de Janeiro, RJ, Brazil

Corresponding author
Rodrigo R. Maia,
rodrigo-maia@uiowa.edu

## ABSTRACT

**Purpose.** This study tested the null hypothesis that different classes of direct restorative dental materials: silorane-based resin, low-shrinkage and conventional (non-flowable and flowable) resin-based composite (RBC) do not differ from each other with regard to polymerization shrinkage, depth of cure or microhardness.
**Methods.** 140 RBC samples were fabricated and tested by one calibrated operator. Polymerization shrinkage was measured using a gas pycnometer both before and immediately after curing with 36 J/cm$^2$ light energy density. Depth of cure was determined, using a penetrometer and the Knoop microhardness was tested from the top surface to a depth of 5 mm.
**Results.** Considering polymerization shrinkage, the authors found significant differences ($p < 0.05$) between different materials: non-flowable RBCs showed lower values compared to flowable RBCs, with the silorane-based resin presenting the smallest shrinkage. The low shrinkage flowable composite performed similarly to non-flowable with significant statistical differences compared to the two other flowable RBCs. Regarding to depth of cure, low-shrinkage flowable RBC, were most effective compared to other groups. Microhardness was generally higher for the non-flowable vs. flowable RBCs ($p < 0.05$). However, the values for low-shrinkage flowable did not differ significantly from those of non-flowable, but were significantly higher than those of the other flowable RBCs.
**Clinical Significance.** RBCs have undergone many modifications as they have evolved and represent the most relevant restorative materials in today's dental practice. This study of low-shrinkage RBCs, conventional RBCs (non-flowable and flowable) and silorane-based composite—by *in vitro* evaluation of volumetric shrinkage, depth of cure and microhardness—reveals that although filler content is an important determinant of polymerization shrinkage, it is not the only variable that affects properties of materials that were tested in this study.

## INTRODUCTION

When dental resin-based composite (RBC) is light cured, stresses develop as a result of the polymerization contraction that accompanies setting, and they may be transferred to the bonded margins of the restoration (*Braga, Hilton & Ferracane, 2003*; *Braga, Ballester & Ferracane, 2005*; *Braga & Ferracane, 2004*; *Ferracane, 2008*; *Asmussen, 1985*). The magnitude of these potentially damaging stresses is a function of certain characteristics of the material, such as its composition (particularly the filler concentration), the reaction kinetics and the degree of conversion of the polymeric matrix (*Braga, Hilton & Ferracane, 2003*; *Braga, Ballester & Ferracane, 2005*; *Braga & Ferracane, 2004*; *Ferracane, 2008*).

The filler content of each RBC is directly related to the mechanical properties and wear resistance of the polymerized product. High volume (%) of different fillers are fundamental to minimizing shrinkage of the composite during polymerization (*Asmussen, 1982*). As the filler content influences both the elastic modulus and volumetric shrinkage, the amount of filler present in an RBC is a major determinant of polymerization contraction stress (*Bayne et al., 1998*), which ultimately affects the integrity of the restoration margin (*Braga, Hilton & Ferracane, 2003*; *Braga, Ballester & Ferracane, 2005*; *Braga & Ferracane, 2004*).

Flowable RBCs differ from their conventional ("non-flowable") counterparts in that they contain substantially less (as much as 25% by weight) filler than conventional RBC (*Labella et al., 1999*), and several studies have shown significant differences in the elastic modulus and volumetric shrinkage between materials of these two classes (*Braga, Ballester & Ferracane, 2005*; *Furuse et al., 2008*). Although the high volumetric shrinkage that characterizes flowable composite materials may lead to high stress values, it is possible that their low elastic modulus could compensate to some degree for the stress buildup, by helping to maintain the marginal seal and integrity of the restoration (*Braga, Ballester & Ferracane, 2005*). Although flowable RBCs generally have a lower elastic modulus than their non-flowable counterparts, in some cases the elastic modulus may not be low enough to provide significant stress relief, as has been observed in studies evaluating unfilled resins (*Braga, Hilton & Ferracane, 2003*).

Efforts to overcome clinical deficiencies of RBCs have led to the development of new matrix materials (*Guggenberger & Weinmann, 2000*). Siloranes have been suggested as alternatives to methacrylates as components of the RBC polymer matrix, due to their hydrophobicity and low polymerization shrinkage (*Weinmann, Thalacker & Guggenberger, 2005*; *Palin et al., 2005*). Siloranes are hybrid systems that contain both silorane and oxirane-based monomers. The individual components of the base resin silorane combined provide two main advantages: low polymerization shrinkage, due to ring opening of the oxirane monomer; and increased hydrophobicity, due to the nature of siloxane species. This system compensates for contraction-induced stress by opening of the oxirane ring during polymerization. The advantage of the hydrophobicity of this restorative material is that it leads to lower absorption of pigments present in the diet, and may reduce the potential for the adhesion of oral biofilms (*Palin et al., 2005*). Additionally, silorane monomers produce RBC systems with better biocompatibility and margin integrity, as

well as lower water absorption and solubility relative to methacrylate-based RBCs (*DeWald & Ferracane, 1987*).

The aim of this study was to measure and compare polymerization shrinkage, depth of cure, and Knoop microhardness (KHN) among low-shrinkage to conventional (non-flowable and flowable) RBCs. The tested hypotheses are that: Silorane and low-shrinkage RBCs will present lower polymerization shrinkage; overall shrinkage of the conventional flowable and non-flowable RBCs is related indirectly to their filler content volumes; and low-shrinkage RBCs will have the greatest depth of cure. Therefore, this *in vitro* study tested the null hypothesis that different restorative materials: low-shrinkage, conventional (non-flowable and flowable) RBCs and silorane not differ from each other with regard to polymerization shrinkage, depth of cure and microhardness.

## METHODS AND MATERIALS

### Materials selection and specimen preparation

In this study, seven restorative dental materials (Table 1) of A2/U shade were selected to minimize the effects of colorants on the light polymerization. All samples were fabricated and tested by one calibrated operator. Materials were evaluated for percentage of filler volume and matrix monomer variation within the major categories of restorative RBC: conventional non-flowable (C), flowable (F) or low-contraction (L). Regarding to the material type based on the filler size, two different groups are present in this study: Nanofilled and Mycro-hybrid RBCs.

### Polymerization shrinkage

Ten samples per group ($n = 70$) were fabricated by placing the material in a 4 mm diameter by 2 mm height stainless steel molds. After the molds were filled, they were placed into a calibrated gas pycnometer AccuPyc$^{TM}$ 1340, Micromeritics$^{®}$, and the volume was measured before and after light curing. Accuracy was ensured by measuring the volume of each specimen five times. Photopolymerization was performed by using a glass slide (2 mm thickness) on top of the mold to support the polywave LED tip (Ultra-Lume LED5 at 600 mW/cm$^2$; Ultradent, South Jordan, Utah, USA) delivering 36 J/cm$^2$ (600 mW/cm$^2$ as measured with a LED radiometer 910726; Kerr, Orange, California, USA) of light energy to each specimen to ensure that all brands and ranges of materials were completely cured.

The polymerization shrinkage was calculated using the equation:

$$PS = \frac{V_i - V_f}{V_i} \times 100$$

where PS is the polymerization shrinkage (in %), $V_i$ is the volume of unpolymerized RBC and $V_f$ is the volume of polymerized RBC.

### Depth of cure

There is disagreement over the best depth of cure evaluation for RBCs. Among the available tests, those assessing the degree of conversion, microhardness and scraping are the most reliable (*Leprince et al., 2012*). Independent of the test used, the depth of cure needs to take

Maia et al. (2015), *PeerJ*, DOI 10.7717/peerj.864

**Table 1  Materials used in this study.**

| Material type/ Commercial name | Type | Matrix type | Photoinitiator system | Filler type | Filler loading (vol%) | Shade | Manufacturer | Batch # |
|---|---|---|---|---|---|---|---|---|
| Mycro-hybrid SureFil® SDR™ Flow (SDR) | F, L | Polymerization modulator, dimethacrylate resins, UDMA | CQ | Ba-B-F-Al silicate glass, SiO2, Sr–Al silicate glass, TiO2 | 44 | U | Dentsply | 91130 |
| Mycro-hybrid Tetric N Flow (TNF) | F | Bis-GMA, Bis-EMA, UDMA, TEGDMA | CQ | Barium glass, ytterbium trifluoride, Ba-Al-fluorosilicate glass, $SiO_2$ | 39 | $A_2$ | Ivoclar /Vivadent | L40758 |
| Nanofilled Filtek Z350 Flow (FZ350F) | F | Bis-GMA, Bis-EMA, TEGDMA | CQ | Agregated zirconia/silica cluster | 55 | $A_2$ | 3M Espe | 1027100529 |
| Mycro-hybrid Esthet-X HD (EXHD) | C | Bis-GMA, Bis-EMA, TEGDMA | CQ | Barium fluoroborosilicate glass and silica | 60 | $A_2$ | Dentsply | L58656 |
| Mycro-hybrid Tetric N Ceram (TNC) | C | Bis-GMA, Bis-EMA, UDMA | CQ | Barium glass, ytterbium trifluoride, Ba-Al-fluorosilicate glass, SiO2 | 55–57 | $A_2$ | Ivoclar /Vivadent | 026700190 |
| Nanofilled Filtek Z350 XT (FZ350) | C | Bis-GMA, Bis-EMA, UDMA, TEGDMA | CQ | Agregated zirconia/silica cluster | 63.3 | $A_2$ E | 3M Espe | 1026600561 |
| Mycro-hybrid Filtek P90 (FP90) | C, L | 3,4-Epoxycyclohexyl ethylcyclopolymethylsiloxane, | CQ, iodonium salt and electron donor | Silanized quartz; yttriumfluoride | 55 | $A_2$ | 3M Espe | 3480370 |

Notes.

F, flowable; C, conventional; L, low-contraction; Bis-GMA, bisphenol-glycidyl-methacrylate; Bis-EMA, bisphenol-a-ethoxydimethacrylate; UDMA, urethane-dimethacrylate; TEGDMA, triethyleneglycoldimethacrylate; HEMA, hydroxyethylmethacrylate; CQ, camphorquinone.

into account the depth at which the transition between the glassy and rubbery state of the resin matrix occurs (*Harrington & Wilson, 1993*).

The depth of cure was determined using a circular stainless steel split mold (6 mm diameter by 5 mm height). Ten samples per group ($n = 70$) were prepared by using the same light curing unit and the amount of energy described previously. A Microtester (Model No. 4206; Instrom Corporation, Norwood, Massachusetts, USA) was used as a penetrometer, according to the methodology of (*Harrington & Wilson, 1993*; *Lowe, 2010*). Immediately after light curing, the molds were inverted such that the unexposed surface (bottom) faced the penetration needle. Pulses of a 12.5N force (1,250 g) were applied using a 0.5 mm diameter needle, at a rate of 1 mm/min, to the middle of the bottom, and the depth of penetration was measured digitally at this point. Depth of cure was calculated using the formula: Depth of cure = Depth of mold—Depth of penetration.

### Knoop microhardness (KHN)

After depth of cure was measured, the same specimens ($n = 70$) were subjected to testing of KHN using a Digital Microhardness Tester (Model no. MMT-X7; Matsuzawa Co., Ltd., Toshima, Kawabe, Japan). The top, light-exposed surface of each specimen was placed directly below the Knoop diamond indenter, and a 500 g load was applied using the indenter, with a dwell time of 15 s. The indentation on the top surface was measured at $100\times$ magnification. The KHN corresponding to each indentation was computed by measuring the dimensions of the indentation and using the formula KHN $= 14.2 \times (F/d^2)$, where $F$ = test load in Newtons; $d$ = longer diagonal of an indentation (in mm). After determining the KHN at the top surface, the split stainless steel mold was opened and KHN values of the side surfaces of the RBC specimens were measured, at 1 mm intervals and working from the top surface down to the level determined as the depth of cure of the RBC sample, using the testing parameters described above. The bottom value for KHN was then recorded.

### Statistical analysis

Statistical analysis was performed using a one-way analysis of variance (ANOVA) and a post-hoc test of Student-Newman-Keuls (SNK) to segregate the materials into groups of similar behavior, and 0.05 was considered the cutoff for significance.

### RESULTS

The results obtained in the present study are displayed in Table 2. Included are mean values ($p < 0.05$) and for the degree of polymerization shrinkage, depth of cure and KHN for each RBC. One-way ANOVA indicated that in each test at least one RBC produced statistically significant differences ($p < 0.05$) from the others.

In regard to:

(1) Polymerization shrinkage, the statistical analysis for the seven composite resins revealed statistically significant differences. FP90 (which is based on the resin silorane) showed the lowest value for shrinkage, followed by the non-flowable RBCs (Tetric N

**Table 2  Arithmetical mean values of all tests (SD).**

| Material ($n = 10$) | Degree of polymerization shrinkage (%) | Depth of cure (mm) | Knoop microhardness (KHN) | | |
|---|---|---|---|---|---|
| | | | Top | Bottom | Reduction (%) |
| SDR | 2.906 (0.04)[E] | 3.071 (0.05)[C] | 72.725 (1.24)[D] | 64.810 (0.04)[G] | 10.37 |
| TNF | 4.217 (0.08)[G] | 2.893 (0.07)[B] | 55.599 (0.02)[B] | 41.858 (0.55)[A] | 24.64 |
| FZ350F | 4.112 (0.05)[F] | 2.837 (0.13)[B] | 53.712 (1.32)[A] | 45.124 (0.16)[B] | 14.09 |
| EXHD | 2.256 (0.09)[D] | 2.612 (0.10)[A] | 77.422 (1.25)[F] | 61.321 (0.53)[D] | 21.01 |
| TNC | 2.031 (0.13)[B] | 2.544 (0.23)[A] | 64.130 (1.15)[C] | 52.029 (0.44)[C] | 18.62 |
| FZ350 | 2.134 (0.07)[C] | 2.567 (0.13)[A] | 78.664 (0.68)[G] | 63.282 (0.81)[F] | 19.89 |
| FP90 | 1.015 (0.12)[A] | 2.679 (0.06)[A] | 73.704 (0.61)[E] | 62.620 (0.69)[E] | 14.98 |

**Notes.**

Values in each column represent the means and standard deviation (in parentheses). Upper-case letters in superscript designate groups whose $p$ values for a given parameter (polymerization shrinkage, depth of cure or KHN) were not statistically different ($p > 0.05$).

Ceram, Filtek Z350XT and Esthet-X HD). SDR represents an intermediate group, with lower values of shrinkage than the other flowable RBCs (FZ350F and TNF).

(2) Depth of cure, the RBCs fell into three distinct groups. SDR exhibited the highest depth of cure. A group of flowable RBCs formed the second group. The non-flowable RBCs represent the third group.

(3) Knoop microhardness, values for the seven composite resins varied widely. As expected, the highest values for hardness at the top surface were exhibited by the non-flowable RBCs. Moreover, when the KHN values at the bottom were evaluated, SDR had the highest value.

## DISCUSSION

The results obtained in this analysis led to rejection of the stated null hypothesis, with the tested RBCs showing distinct qualities with regard to polymerization shrinkage, depth of cure and microhardness. The composition of an RBC determines its physical properties in polymerized form. In this study, variations in the polymeric matrix and the filler concentration of new RBCs gave rise to mechanical properties that could prove clinically advantageous over those of the conventional, gold-standard RBCs that were tested. RBCs that are characterized by lower shrinkage and greater depth of cure and by similar hardness at both the top and bottom surface could improve on the current bulk-filling techniques.

Given that volumetric shrinkage is directly related to the organic matrix of the composite resin, it was expected that SDR and silorane-based resins would shrink less than conventional methacrylate-based RBC (*Palin et al., 2005*). In addition, the amount of filler particles is related to polymerization shrinkage; non-flowable RBCs, which have more filler than their flowable counterparts, typically shrink less during polymerization than do flowable RBCs (*Braga & Ferracane, 2004*). This emphasis on shrinkage is important; when high, it may contribute to a restoration's failure by affecting the marginal integrity, and possibly also lead to post-operative sensitivity (*Lowe, 2000*). This study corroborates that volumetric shrinkage ascends for the tested materials in the following

order: silorane-based resin, non-flowable RBCs, and flowable RBCs. Nevertheless, SDR presented values of volumetric shrinkage that were very similar to those of non-flowable ones and significantly lower than those for other flowable RBC tested. Its inability to improve on the non-flowable materials with respect to shrinkage may be due to the fact that the low contraction of the resin monomer could not completely compensate for the lower percentage of filler (44%) in this RBC.

Flowable RBCs typically have a greater depth of cure than their non-flowable counterparts. This is because polymerization at depth is directly related to the filler's particle size and dispersion, with smaller size and greater dispersion promoting differences in scattering of the light through the material (*Leprince et al., 2012*). SDR presented statistically significant increase in depth of cure up to 3 mm. This is an improvement over all of the RBCs studied (*Moore et al., 2008*), though it is also less than the 4 mm advertised by the manufacturer (*Asmussen, 1985*). However, other materials also failed to meet the depth-of-cure criteria (above 2 mm thickness). This may be due in part to the fact that depth of cure is influenced by RBC shade.

Knoop microhardness was used as a second method to assess the depth of cure in this study, based on the discovery by *Flury et al. (2012)* and *Salerno et al. (2011)* that for bulk-fill materials the ISO 4049 method overestimated depth of cure compared to its determination by microhardness tests. The evaluation of top and bottom KHN, and of the percentage reduction, revealed that the flowable RBCs generally produced lower levels of microhardness at the top. The exception was SDR, whose top KHN was significantly higher. Regarding bottom-surface KHN, SDR had the highest mean values, regardless of viscosity, among the materials evaluated in this study. Notably, the ratio of the KHN at the top vs. bottom of the specimen was the lowest in the case of SDR. This fact could be related to the higher depth of cure obtained in the present study.

## CONCLUSIONS

The following conclusions may be drawn:

1. The silorane-based resin (FP90) performed as observed in previous studies, exhibiting the least polymerization shrinkage among the RBCs tested here.
2. The low shrinkage flowable composite (SDR) performed similarly to non-flowable with significant difference compared to the other flowable RBCs.
3. All materials tested presented statistical significant differences for microhardness from the top and from the bottom.

### Funding

This research was supported by Dentsply Caulk (Latin America Division grant # year:2011). The funders had no role in study design, data collection and analysis, decision to publish, or preparation of the manuscript.

## Grant Disclosures

The following grant information was disclosed by the authors:
Dentsply Caulk Latin America Division.

## Competing Interests

The authors declare there are no competing interests.

## Author Contributions

- Rodrigo R. Maia conceived and designed the experiments, performed the experiments, wrote the paper, reviewed drafts of the paper.
- Rodrigo S. Reis conceived and designed the experiments, performed the experiments.
- André F.V. Moro, Cesar R. Perez and Bárbara M. Pessôa contributed reagents/materials/analysis tools, prepared figures and/or tables.
- Katia R.H.C. Dias analyzed the data.

## Supplemental Information

Supplemental information for this article can be found online at http://dx.doi.org/10.7717/peerj.864#supplemental-information.

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
