# Peer review of "Properties evaluation of silorane, low-shrinkage, non-flowable and flowable resin-based composites in dentistry"

_PeerJ, doi:10.7717/peerj.864_

## Round 0.1 · original submission · Minor Revisions

· Academic Editor

Minor Revisions

Please address the minor concerns raised by the reviewers. The references citation format is outlined in the author guidelines, available online.

Reviewer 1 ·

Basic reporting

Very interesting study comparing physical characteristics of different dental resin-based composites. The article is well presented and the study well designed.

Experimental design

Well designed study. It would help the reader, however, to know which shade was used and how many specimens were produced (sample size of the study).

Validity of the findings

The conclusions of the study are supported by the result, however, see comments on the Experimental Design.

Additional comments

Well written manuscript, very interesting topic. The manuscript would need minor revisions:
1) Shade of the products;
2) Sample size for each test
3) punctuation at the Reference List. It is missing or inconsistent.

Reviewer 2 ·

Basic reporting

A well done study. Need to simplify the results as explained in the Summary, Abstract and conclusions. Grammatical corrections required.

Experimental design

No Comments

Validity of the findings

No comments

Additional comments

The type of composites rather than the trade names need to be referred.The page numbers of the references are not as per the guidelines.

---

## Round 0.2 · accepted · Accept

· Academic Editor

Accept

The revision has addressed all the comments except a minor typo (see attached annotated PDF). Please correct this in your final proof as indicated.

Reviewer 2 ·

Basic reporting

No Comments

Experimental design

No Comments

Validity of the findings

No Comments

Additional comments

Please check the highlighted areas and do the needful.

Annotated reviews are not available for download in order to protect the identity of reviewers who chose to remain anonymous.